# An Improved HRPE-Based Transcriptional Output Reporter to Detect Hypoxia and Anoxia in Plant Tissue

**DOI:** 10.3390/bios10120197

**Published:** 2020-12-03

**Authors:** Gabriele Panicucci, Sergio Iacopino, Elisa De Meo, Pierdomenico Perata, Daan A. Weits

**Affiliations:** 1Biology Department, University of Pisa, 56126 Pisa, Italy; g.panicucci13@studenti.unipi.it (G.P.); s.iacopino@santannapisa.it (S.I.); 2Institute of Life Sciences, Scuola Superiore Sant’Anna, 56127 Pisa, Italy; pierdomenico.perata@santannapisa.it; 3Department of Pharmacy and Biotechnology, University of Bologna, 40126 Bologna, Italy; elisa.demeo2@studio.unibo.it

**Keywords:** oxygen, hypoxia, ERF-VII, UnaG, anoxia, reporter, HRPE, plants, biosensor, fluorescence

## Abstract

Oxygen levels in plant tissues may vary, depending on metabolism, diffusion barriers, and environmental availability. Current techniques to assess the oxic status of plant cells rely primarily on invasive microoptodes or Clark-type electrodes, which are not optimally suited for experiments that require high spatial and temporal resolution. In this case, a genetically encoded oxygen biosensor is required instead. This article reports the design, test, and optimization of a hypoxia-signaling reporter, based on five-time repeated hypoxia-responsive promoter elements (HRPE) driving the expression of different reporter proteins. Specifically, this study aimed to improve its performance as a reporter of hypoxic conditions by testing the effect of different untranslated regions (UTRs) at the 5′ end of the reporter coding sequence. Next, we characterized an optimized version of the *HRPE* promoter (*HRPE-Ω*) in terms of hypoxia sensitivity and time responsiveness. We also observed that severe oxygen deficiency counteracted the reporter activity due to inhibition of GFP maturation, which requires molecular oxygen. To overcome this limitation, we therefore employed an oxygen-independent UnaG fluorescent protein-coupled to an O_2_-dependent mCherry fluorophore under the control of the optimized *HRPE-Ω* promoter. Remarkably, this sensor, provided a different mCherry/UnaG ratiometric output depending on the externally imposed oxygen concentration, providing a solution to distinguish between different degrees of tissue hypoxia. Moreover, a ubiquitously expressed UnaG-mCherry fusion could be used to image oxygen concentrations directly, albeit at a narrow range. The luminescent and fluorescent hypoxia-reporters described here can readily be used to conduct studies that involve anaerobiosis in plants.

## 1. Introduction

The study of low oxygen (hypoxia) conditions has attracted growing attention in recent years across several fields of biology, including plant science [1,2,3]. In plants, hypoxia is a well-characterized stressful condition associated with submergence or waterlogging, since it restricts respiratory metabolism. Despite its apparent negative consequences for efficient energy production, hypoxia has been observed to occur as a chronic endogenous condition in several different tissues. These include tissues with obvious restriction in gas diffusions, such as tubers, fruits, and seeds, but chronic hypoxia was also recently measured in meristematic tissues [4,5,6]. For example, hypoxia-induced gene expression was shown to occur in conjunction with the early stages of lateral root primordia development, and O_2_ microprofiling has revealed a steep oxygen gradient in the shoot apical meristem (SAM) [7,8]. In these tissues, hypoxia acts to restrict proteasomal degradation of N-degron pathway substrates, which therefore accumulate in meristematic tissue and regulate developmental processes [1,6,9].

Moreover, hypoxia is induced upon infection with several different pathogens, including gall producing *Plasmodiophora* and *Agrobacterium*, and necrotrophic *Botrytis cinerea* fungus [10,11,12]. Thus, hypoxic conditions occur more broadly than previously considered, which highlights the importance of robust and easy-to-use oxygen biosensors [13]. Currently, the state-of-the-art technique for tissue oxygen measurements is through the use of optodes or Clark electrodes, which are both produced commercially as miniaturized microsensors with a minimal tip diameter of 50 µm and 3–5 µm, respectively. While the remarkably small size of Cark electrodes provided the possibility to measure small tissues, including meristems, these sensors do have some drawbacks [14]. Namely, the use of microsensors for continued live imaging may be limited if their insertion allows O_2_ to diffuse into the tissue. Miniaturized microsensors are minimally invasive and confer high spatial resolution, but they are also more fragile, due to a thinner protective glass casing, which increases the risk of breaking and reduces their ability to penetrate hard tissue. Instead, fluorescence-based biosensors can be imaged with minimal damage to the tissue and, when photoxicity from excitation light is kept to a minimum, can be used for live imaging.

Iacopino and colleagues employed a synthetic oxygen sensor based on the animal prolyl-4-hydroxylase domain (PHD) enzymes oxygen signaling machinery in plants [15]. This molecular device showed an activation upon a decrease in the oxygen concentration to 5% O_2_ in a reversible manner. Moreover, this oxygen sensor only showed O_2_-responsiveness in the presence of PHD and was not affected by ABA, cold, or salt stress, which can activate the expression of some anaerobic genes. Previous reports also show successful usage of hypoxia-responsive promoters to detect hypoxia within the tissue [8]. In the case of the SAM, galls, and fungal induced lesions, the presence of hypoxia was confirmed using oxygen microsensors, establishing the functionality of these hypoxia-signaling reporters to predict underlying low oxygen conditions [7,10,12]. However, the O_2_-dependency of GFP maturation limits the use of current hypoxia reporters under strongly limiting oxygen concentrations.

Since the discovery and characterization of green fluorescent protein (GFP) from *Aequorea victoria*, an enormous number of fluorophores have been discovered or generated through mutagenesis [16]. However, while the diversity of these fluorophores covers virtually the entire spectrum of visible and near-infrared light, nearly all fluorescent proteins require an O_2_-dependent oxidation step of the chromophore during the maturation of the protein [17]. This restricts their applicability under strongly limited oxygen conditions, since de novo produced protein does not fluoresce without oxygen. As an answer to this problem, few O_2_-independent fluorescent proteins have been identified. These include the flavin binding fluorescent proteins (FbFP) and a fluorescent protein (FP) isolated from *Anguilla japonica*, termed UnaG, which requires bilirubin for fluorescence via high-affinity noncovalent binding [18,19,20]. UnaG is a bright green fluorescent protein that has a low pKa making it suitable for hypoxia imaging. However, UnaG is sensitive to photobleaching, due to oxidation of its cofactor bilirubin by 488 nm excitation light [21]. Employing its O_2_-independent fluorescence, UnaG was previously used to image hypoxia in *E.coli* and mouse glioblastoma tumors [22,23].

To detect underlying hypoxic conditions in the SAM, a synthetic promoter based on a five-time repeat of the hypoxia-responsive promoter element (HRPE) fused to GUS-GFP was characterized and used to image hypoxic responses in the shoot apical meristem [7]. The HRPE element was previously identified as a major cis element bound by the ethylene response factors (ERF) group VII [24], which are essential transcriptional regulators of anaerobic gene expression [25,26,27]. ERF-VII proteins only accumulate in the nucleus upon hypoxia, due to the activity of plant cysteine oxidases (PCO), which use O_2_ as a co-substrate to oxidize N-terminally exposed cysteine [28,29,30]. This post-translational modification labels cysteine-initiating proteins for further modifications, which leads to their degradation via the N-degron pathway [31]. Thus, the HRPE element confers hypoxia-inducibility in an ERF-VII dependent manner [32]. Here, we improved and characterized hypoxia reporters based on a five-time repeat of the HRPE element through the combination of O_2_-dependent and independent reporter proteins, providing a means to detect hypoxia and anoxia in plant tissue.

## 2. Materials and Methods

### 2.1. Plant Materials

*Arabidopsis thaliana* Columbia-0 seeds were used as a wild-type ecotype. The *HRPE:GUS-GFP* lines were previously described [7]. The *HRPE-Ω:GUS-GFP* and *HRPE-ADH:GUS-GFP* lines were newly generated as described in the cloning section. Wildtype *Nicotiana benthamiana* was used for transient transfection.

### 2.2. Plants Growth Conditions

*Arabidopsis thaliana* seeds for in vitro cultivation were sown on half-strength, agarized Murashige and Skoog medium and stratified for 48 h at 4 °C in the dark. Seeds were then germinated in short-day conditions (12 h light 12 h dark, 23 °C, 50% relative humidity, 100 µmol/m^2^/s light intensity). Seven-days old seedlings were used for GUS staining and GFP imaging following hypoxic treatment. *Nicotiana benthamiana* seeds were germinated on moisturized filter paper and then transferred to a soil-perlite mixture (3:1 ratio). Plants were then grown in long-day conditions (18 h light, 6 h dark, 23 °C, 50% relative humidity, 100 µmol/m^2^/s light intensity). Leaves of four-weeks old plants were used for transient transfection.

### 2.3. Hypoxia Treatments

Hypoxia treatments were performed by placing the plants in a Gloveless Anaerobic chamber (COY). Mixing of nitrogen gas and atmospheric air was performed to reach the indicated oxygen concentration for each experiment. A Pyroscience FireStingO2 (FSO2-2) oxygen meter, together with OXSP5 sensor spots, were used to confirm the desired oxygen concentration inside the glovebox.

### 2.4. Constructs Cloning and Assembling

HRPE promoter variants (Appendix A) were de novo synthesized as gateway entry vectors by GeneArt service (Thermo-Fisher Scientific, Waltham, MA, USA). Starting from the previously described Hypoxia-responsive promoter element (HRPE) [7], we introduced the 5′-leader sequence (called Ω) of tobacco mosaic virus (TMV) [33] downstream of the HRPE promoter, generating the *HRPE-Ω*. Similarly, the 5′ UTR (-254 upstream of the ATG) sequence of *At1g77120* (*ADH1*) was placed downstream of the HRPE promoter, generating the *HRPE-ADH*. Both promoter units were designed flanked by attL sites for subsequent application in gateway cloning. For transient experiments, HRPE entry vectors were recombined in the pGreen800GW destination vector [34,35] by gateway cloning. The *HRPE-Ω:GG* (GUS-GFP) and *HRPE-ADH:GG* constructs were realized by gateway cloning using the pH7GWFS7 [36] destination vector.

Transcriptional units encoding the Pp2FbFP, iLov, and UnaG fluorophores (Appendix A) were designed as DNA strings carrying an additional CACC sequence at the 5′-end for immediate subcloning into the pENTR/D-TOPO^®^ vector (Thermo-Fisher Scientific). For protoplast transfection, all entry fluorophore vectors were recombined into the p2GW7 [36] destination vector using gateway cloning. Gateway reactions were performed using the Gateway™ LR Clonase™ II Enzyme mix (Thermo-Fisher Scientific).

The *HRPE-Ω*:UnaG-mCherry and *pUBQ10:UnaG-mCherry* constructs were realized by GreenGate cloning [37]. The *HRPE-Ω* promoter was amplified with ggHRPE-Ω_Fw (aacaGGTCTCaACCTGCCGCCCCCTTCACC) and ggHRPE-Ω_Rv (aacaGGTCTCaTGTTCCCTTTCGACTAGAA) primers and cloned into the T_0_ GreenGate pGGA000 entry module using BsaI restriction sites. The UnaG transcriptional unit was amplified with ggUnaG_Fw (aacaGGTCTCaGGCTccATGGTCGAGAAGTTCG) and ggUnaG_Rv (aacaGGTCTCaCTGATTCAGTAGCACGTCTG) and cloned into the T_0_ GreenGate pGGC000 entry module using BsaI restriction sites. The GreenGate reaction, containing 100 ng of HRPE-Ω or pGGA006 (UBQ10) (promoter), pGGB003 (dummy N-tag), UnaG (CDS), pGGD003 (mCherry C-terminal fusion tag), pGGE009 (UBQ10 terminator), pGGF005 (Hygromycin resistance cassette), and pGGZ001 (destination vector) entry modules, 1 µL of BsaI fast digest (Thermo-Fisher Scientific) and 2.5 µL of Anza™ T4 DNA Ligase Master Mix (Thermo-Fisher Scientific) in a final volume of 10 µL, was performed in a thermocycler with 30 cycles of 37 °C for 2 min and 16 °C for 2 min, followed by 50 °C for 5 min and 80 °C for 5 min. Destination vectors were tested by restriction and sequencing to confirm the correct insertion of each module.

### 2.5. Fluorescence Microscopy

Seven-days old seedlings were kept for 16 h at either 1%, 2.5%, or 21% *v/v* oxygen concentration and then used for GFP imaging. Imaging was performed with a Leica THUNDER imager model organism using bandpass filters for GFP (excitation: 470/40 nm, emission: 525/50 nm) and RFP (excitation: 546/10 nm, emission: 605/70 nm). Confocal laser scanning microscopy was performed using a Zeiss airyscan 800. Fiji was used to quantify UnaG and mCherry fluorescence intensity [38]. Each data point represents the average mCherry/UnaG ratio at the nuclei and cytosol.

### 2.6. Statistical Tests and Data Representation

One and two-way analysis of variance (ANOVA) were performed using GraphPad Prism 7.0. Boxplot limits represent the 25th and 75th percentiles of each dataset. The whiskers extend to the lowest and highest data point. The central line represents the median. Histograms represent the mean ± the standard deviation.

### 2.7. Histochemical GUS Staining

Histochemical GUS staining of seedlings expressing the HRPE variants fused to GUS-GFP, was performed by four hours or overnight incubation with GUS staining solution (100 mM buffer phosphate, 0.1% Triton X-100, EDTA pH 8 10 mM, potassium ferrocyanide 0.5 mM, potassium ferricyanide 0.5 mM, X-Gluc 200 mM) and then cleared in several washes of 70% (*v*/*v*) ethanol. Images were taken using the THUNDER imager model organism (Leica microsystems).

### 2.8. RT-qPCR

To assess mRNA levels of *GFP*, *GUS*, and *PCO1* (*At5g15120*), seven-day old *Arabidopsis thaliana* seedlings were grown vertically on agarized half-strength MS plates. Full seedlings were harvested after four hours of 21% or 0% O_2_ treatments. Total RNA extraction, subsequent DNase treatment, cDNA synthesis, and RT–qPCR analysis were performed as described previously [28].

### 2.9. Transient Transfection of Nicotiana Benthamiana

Leaves of four-weeks old *Nicotiana benthamiana* were used for agroinfiltration and transient transfection. *Agrobacterium tumefaciens* cultures (strain GV3101) were grown overnight in LB media, selection antibiotics (50 ug/mL), and 20 µM acetosyringone. The cultures were then pelleted, and the bacteria were resuspended in a 200 µM acetosyringone MMA solution (MS 5 g/L, MES 1.95 g/L, sucrose 20 g/L, pH 5.6), reaching a final OD600 of 0.4. The abaxial sides of the third, fourth, and fifth leaves of *Nicotiana benthamiana* plants were infiltrated with the Agrobacterium solution using a 5 mL syringe. Following infiltration, plants were kept in the dark for 4 h and then moved to standard long day growing conditions as described before. Disks cut out of infiltrated leaves were used after 48 h for confocal imaging or protein extraction.

### 2.10. Reporter Activity Assay

Four-weeks old *Nicotiana benthamiana* plants were grown for 48 h in long-day conditions after agroinfiltration. Leaves were then cut into disks, placed into 6-well plates filled with water and subjected to hypoxic treatment. Following treatment, leaf disks were harvested in liquid nitrogen and used for protein extraction following the Dual Luciferase Reporter (DLR) Assay System (Promega, Madison, WI, USA). Disks were ground in 400 µL Passive Lysis Buffer and then diluted 1:300 in the same buffer. Samples were vortexed, and 6 µL of protein extract was used for DLR assay. We used 30 µL of Luciferase Assay ReagentII (Promega) for firefly luciferase activity, and 30 µL of Stop and Glo (Promega) to quench firefly activity and induce renilla luciferase activity. All luciferase activity readings were performed using a Lumat LB 9507 luminometer (Berthold Technologies, Oak Ridge, TN, USA).

## 3. Results

### 3.1. Optimisation and Characterization of HRPE-Based Hypoxia Reporters

We first set out to improve the dynamic range of the original HRPE-based reporter by improving the translation of its reporter protein. Previous reports highlighted how hypoxia generally inhibits the plant translation machinery, due to ATP shortage, while specific proteins are still selectively produced [36,37]. Thus, we reasoned that we could improve the translation of reporter genes under oxygen limitation by including the untranslated region (UTR) present at the 5′ of hypoxia-inducible mRNAs or using a strong viral UTR. We selected the 5′ UTR of *Arabidopsis thaliana* alcohol dehydrogenase (*ADH1*, *AT1g77120*) mRNA and the Ω-leader region present in the 35S promoter of the Tobacco Mosaic Virus (CaMV). We fused either new version of the *HRPE* promoters to a firefly luciferase (FLUC) reporter (*HRPE-ADH:FLUC* and *HRPE-Ω:FLUC*) and used a renilla luciferase (RLUC) driven by a 35S CaMV promoter as normalization control (Figure 1a). When transiently transfected in *Nicotiana benthamiana* leaves, the *HRPE-Ω:FLUC* showed a stronger increase in luminescence signal after five hours of hypoxia treatment, as compared to the original *HRPE* or the *HRPE* fused to the 5′ UTR of *ADH1* (Figure 1b).

Next, we characterized the O_2_-sensitivity and the response-time of the *HRPE-Ω* construct. *HRPE-Ω:FLUC* signal was observed within two hours of hypoxia treatment, but was highest after four hours (Figure 1c). Significantly increased *HRPE-Ω:FLUC* luminescence was observed progressively at O_2_ concentrations of 5% and 1% *v/v* in air, while 10% O_2_ showed a mild induction, although undetectable by parametric statistics (Figure 1d). This indicates that the strength of hypoxia-responses in plants depends on the length and severity of the hypoxic condition. Taken together, these data demonstrate that the *HRPE-Ω* reporter can be applied as a hypoxia signaling output reporter within two hours of treatment and is activated at a range of 0–5% O_2_.

### 3.2. Stable in Planta Expression of HRPE Reporters

We stably introduced the *HRPE* variants driving a chimeric GUS-GFP reporter protein (*HRPE:GG*, Figure 2a) in *Arabidopsis thaliana* plants. Strong overnight hypoxia (1% O_2_) treatments led to an activation of HRPE promoter activity in all tissues, while a milder treatment (2.5% O_2_) induced GUS activity primarily in the shoot apex, young primordia, and in the root (Figure 2b,c). In aerobic conditions, GUS staining was observed in the shoot apex of each reporter line, although only for *HRPE-ADH* and the original *HRPE* variant when GUS staining was performed overnight (Figure 2b, Appendix A). This is in line with the hypoxic status of this tissue [7]. Confocal microscopy imaging of GFP in 7-day old root tips of *HRPE-Ω:GG* plants showed that this tissue does not activate *HRPE* in aerobic conditions, while hypoxia led to the induction of GFP signal (Figure 2c). Remarkably, patchy patterns of green fluorescence were observed in hypoxic root tips, and a comparable signal was observed in *HRPE-ADH:GG* and the original *HRPE:GG* lines, indicating that it is not an artifact induced by the omega 5′ UTR or the genomic position of the transgene (Figure 2d).

To investigate the activity of *HRPE-Ω:GG* under anoxia, we shortened the treatment time to four hours to avoid cell death. Anoxia treatments did not lead to an increase in GFP in root tips, and GUS staining revealed heterogenicity in *HRPE-Ω:GG* reporter activity under this condition (Figure 2e,f). RT-qPCR analysis of *GUS* and *GFP* transcripts in *HRPE-Ω:GG* plants revealed a strong increase in *GUS* and *GFP* mRNA upon anoxia, which was comparable to the induction of the endogenous hypoxia-inducible *PCO1* transcript (Figure 2g). Therefore, while the *HRPE-Ω* shows robust activation upon anoxia, the aberrant induction of GUS activity and GFP fluorescence at anoxia hints at impaired translation or maturation of the reporter.

### 3.3. Generation of Anoxia Sensors

The O_2_-dependent maturation of GFP limits the usage of *HRPE-Ω:GG* to conditions at which the reporter is induced (>5% O_2_), but also sufficient oxygen is available for GFP to fluoresce. Indeed, while anoxia treatments permitted variable, but detectable GUS activity, GFP fluorescence was completely impeded when driven by the *HRPE-Ω* (Figure 2e,f). Moreover, it is plausible that GFP fluorescence is at least partially affected at hypoxic conditions, leading to an under-appreciation of reporter activity. To circumvent this drawback of GFP, we investigated the possibility of using O_2_ independent fluorophores, which have been characterized in vitro and in vivo in metazoans or bacteria [18,19]. Among these are the flavin mononucleotide binding cyan-green fluorophores FbFP, iLOV, and the bilirubin dependent green fluorescent UnaG protein. To test their potential application in plants, we first observed their fluorescent signal in transiently transformed *Arabidopsis thaliana* mesophyll protoplasts using a constitutive *35S:* promoter. In protoplasts, detectable fluorescence was observed for Pp2FbFP and UnaG, but not for iLOV (Figure 3a). Among these, UnaG showed the strongest fluorescence, which matches reports from publicly available databases (Fpbase.com).

Next, we generated hypoxia and anoxia reporters based on *HRPE-Ω* driving a fusion of UnaG and mCherry (*HRPE-Ω:UnaG-mCherry*, Figure 3a). mCherry requires oxygen for maturation, and therefore, should not fluoresce under anoxic conditions, which instead does not impair UnaG fluorescence. In this manner, we expected to distinguish between tissue anoxia, which should lead to exclusively green UnaG fluorescence, and moderately hypoxic tissue, showing green and orange fluorescence. To test this hypothesis, *HRPE-Ω:UnaG-mCherry* was transiently transformed in *Nicotiana benthamiana* leaves, and treated with different oxygen concentrations. At oxygen concentrations of 2% and 5% we could observe UnaG and mCherry fluorescence, while no signal was detected under 21% O_2_, confirming that the *HRPE-Ω:UnaG-mCherry* reporter drives expression in response to hypoxia (Figure 3c). In line with the requirement of oxygen for the maturation of mCherry, a lower ratio of mCherry/UnaG was observed at 2% versus 5% O_2_, while anoxic treatment only led to UnaG fluorescence and no detectable mCherry signal (Figure 3c,d). This shows that the *HRPE-Ω:UnaG-mCherry* variant can be used to detect tissue anoxia, while its ratiometric UnaG/mCherry output can be used to infer the actual O_2_ concentration. Based on these observations, we reasoned that a direct, and hypoxia signaling independent, O_2_ sensor could be generated using the same UnaG-mCherry fluorescent pair, but employing a constitutive *UBQ10* promoter. Remarkably, a linear relationship between the mCherry/UnaG ratio and the O_2_ concentration was found at 0.5–5% O_2_ (Figure 4a,b). Instead, no significant difference was observed between 5% and 21% O_2_. Therefore, the ratiometric output of *pUBQ10:UnaG-mCherry* can be used to detect hypoxia, but it is not suitable for imaging of the oxic status of well-oxygenated tissue.

## 4. Discussion

In this article, we reported the optimization and characterization of a hypoxia signaling responsive reporter, which was used to detect hypoxia and anoxia in vivo. Fusing a 5′ Ω-UTR to the end of the five-times repeat of the HRPE sequence improved the signal output under hypoxia, while the use of the *ADH1* 5′ UTR did not. The latter was unexpected since hypoxic conditions significantly induce *ADH1*, and its mRNA is known to be selectively translated at such conditions [36]. The increased hypoxia output conferred by the Ω-UTR likely represents a constitutive increase in translation efficiency, which may be able to overcome the general downregulation of translation associated with hypoxia [36]. The *HRPE-Ω* promoter sequence was found to drive expression of a reporter within two hours from the onset of hypoxia. This is slightly delayed compared with previous reports where hypoxia-responsive transcripts were found to be significantly increased within one hour of treatment [39,40]. This lag may represent the time required for translation and folding of the FLUC reporter protein after the onset of hypoxia, or it may hint at a different responsiveness of *Nicotiana benthamiana* as compared to *Arabidopsis thaliana*. The time required to observe a detectable signal of *HRPE-Ω* may be decreased through the use of a NanoLuc luminescent reporter, which is of smaller size and brighter, compared to FLUC [41]. The selection of a more rapidly maturing fluorophore is also a valid alternative to tackle this aspect. An extensive analysis of a collection of commonly used fluorophores revealed that while eGFP belongs to the rapidly maturating fluorophores and shows 90% fluorescence intensity within 62.8 ± 6.6 min at 32 °C, mGFPmut3d achieves this almost four-times as fast [42].

*HRPE-Ω* was activated when the external O_2_ concentration dropped below 5%, which correlates with the accumulation of nuclear RAP2.12 and likely reflects the affinity of PCOs for oxygen [38,39]. Interestingly, in young *HRPE-Ω:GG* plants, *HRPE-Ω* GUS activity was primarily found at 2.5% O_2_ in the root and shoot apex region, while 1% O_2_ treatment led to an increase of the reporter in all tissues, hinting at a different sensitivity of these tissues to hypoxia. While the SAM is chronically hypoxic, reducing the external oxygen concentration likely accentuates this further, leading to a stronger increase in *HRPE-Ω*:GG activity, compared to the rest of the shoot [7]. Roots are more likely to experience hypoxic conditions, due to waterlogging, and may, therefore, respond more sensitively to mild hypoxic conditions. Curiously, hypoxia led to a patchy GFP signal in the root tip with significant differences in GFP intensity between cells. Although the distribution of the GFP signal appeared random rather than following a defined pattern or a gradient, this may hint at a different sensitivity to hypoxia of cells or at altered efficiency of GFP translation, depending on their state within the cell cycle [43].

Almost all genetically encoded biosensors rely on fluorescent proteins that undergo an oxygen-dependent maturation step to fluoresce, and this prevents their utilization under strongly oxygen limiting conditions. Similarly, the catalysis of luciferin requires oxygen, and therefore, in vivo analysis of *pHRPE:FLUC* activity, i.e., by spraying plants with luciferin, under strong oxygen limiting conditions, is not expected to result in detectable reporter activity. Here, we tested a previously identified O_2_-independent fluorescent protein, UnaG, and found that it was able to produce detectable fluorescence in plants subjected to anoxic conditions. This enables its use as reporter proteins in plants, in particular for the study of chronically hypoxic tissues, such as meristems and galls. Indeed, whilst mCherry was found to display reduced fluorescence at <2% O_2,_ the signal emitted by UnaG was confirmed as O_2_-independent (Figure 3c), indicating that the latter would be a more suitable reporter when performing experiments at O_2_ concentrations below the 2% threshold. Based on these results, we generated an *HRPE-Ω* reporter driving an UnaG-mCherry fusion protein. Indeed, the superiority of this sensor compared to the GFP based one was apparent by its striking UnaG fluorescence under anoxia, compared to no detectable GFP signal. Likewise, no signal for mCherry was observed at 0% O_2_, and mCherry maturation was also negatively affected at 2% O_2_, but not at 5% O_2_, as indicated by a lower ratio of mCherry/UnaG intensity. Thus, while *HRPE-Ω* provides inducibility at 5% O_2_, its ratiometric output can also be employed to infer the actual oxygen concentration. Moreover, when expressing the mCherry/UnaG pair using a constitutive *UBQ10* promoter, we observed a linear relationship between mCherry/UnaG and an O_2_ concentration range of 0.5 to 5%. Therefore, while not suitable as a sensor for moderate to high O_2_ levels, it can be used to detect strong hypoxia when also providing a means to detect spatial differences in the O_2_ concentration within chronic hypoxic tissue, such as meristems.

While the HRPE-based sensors described here should prove as a useful and robust tool to detect hypoxia and anoxia in tissue and to quantify hypoxic responses, one should bear in mind that they act primarily as an output of hypoxia signaling, not actual O_2_ levels. Recent reports found that ethylene, nitric oxide, and ATP levels impact the plant oxygen-sensing pathway, and this should, therefore, be taken into consideration when interpreting hypoxia-signaling output reporters [40,44,45,46]. On the flip side, a combination of the here described HRPE-based reporters together with direct O_2_ concentration measurements and the *pUBQ10:UnaG-mCherry* sensor may be an elegant strategy to disentangle hypoxia signaling from the tissue O_2_ concentration.

## Figures and Tables

**Figure 1 biosensors-10-00197-f001:**
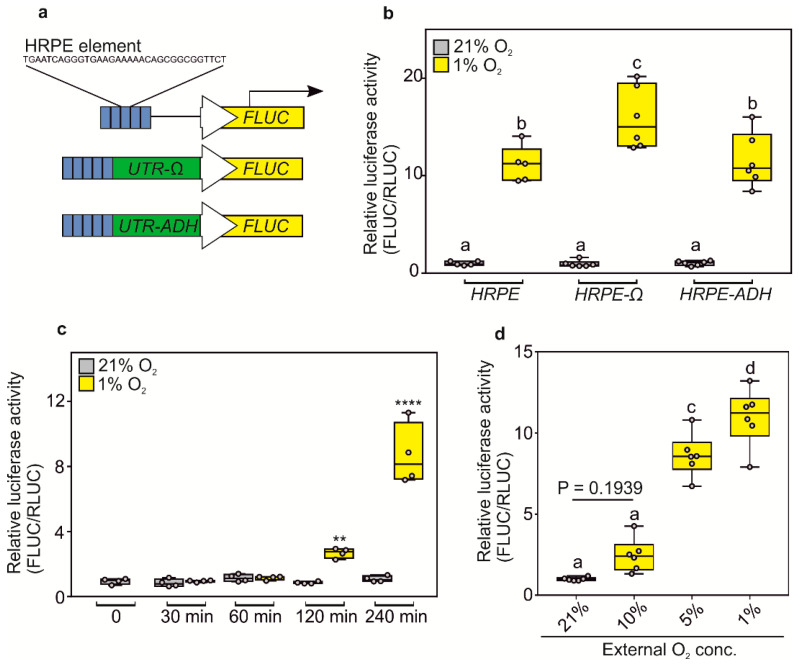
(**a**) Schematic representation of HRPE based hypoxia reporters, which confers the expression of firefly luciferase (FLUC) in low O_2_ conditions. (**b**) Hypoxia-inducibility of different HRPE variants analyzed by transient transactivation assays. 21% and 1% O_2_ treatments were performed for 5 h (**c**) Response-time of *HRPE-Ω* to hypoxia (1% O_2_) or normoxia (21% O_2_) treatment. (**d**) O_2_-concentration responsiveness of *HRPE-Ω*. Treatments were performed for 4 h. All luminescence measurements were carried out using protein extracts isolated from transiently transfected *Nicotiana benthamiana* leaves. *35S:RLUC* was used as a transformation control. For (**b**,**c**) a two-way analysis of variance (ANOVA) was carried out, followed by a Tukey post hoc test. For (**d**), one-way ANOVA followed by Tukey post hoc test. Letters or stars indicate a statistical significant difference (letters, *p* value < 0.05, ** *p* value < 0.01, **** *p* value < 0.0001).

**Figure 2 biosensors-10-00197-f002:**
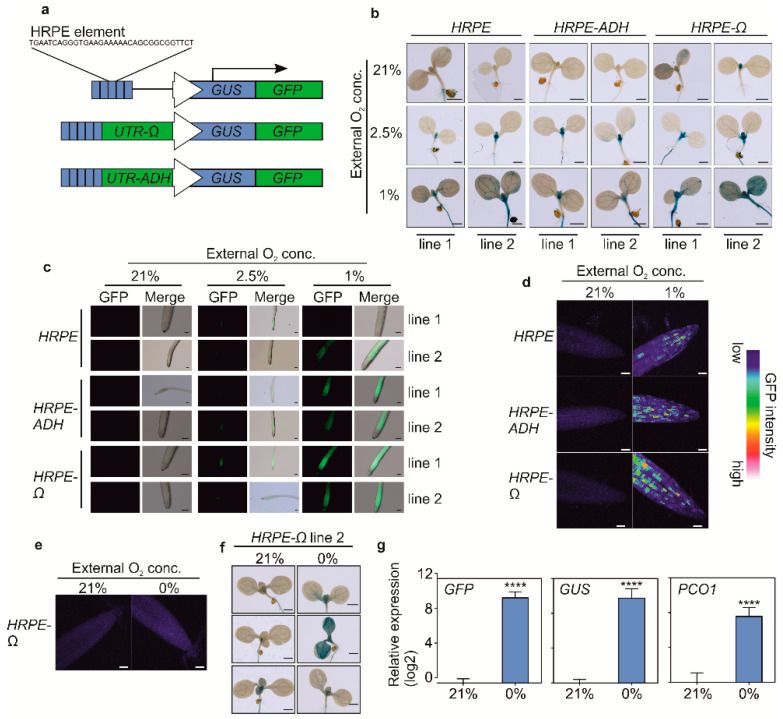
(**a**) Schematic representation of *HRPE:GG* based hypoxia reporters used to transform *Arabidopsis thaliana*. (**b**) Histochemical GUS staining of *HRPE:GG* variants following 21% and 1% O_2_ treatments. Scale bar: 1 mm (**c**) GFP signal in roots of *HRPE*:GG variants following exposure to 21%, 2.5% or 1% O_2_. Scale bar: 100 µm (**d**) Confocal microscopy analysis of GFP intensity in root tips of *HRPE*:GG variants exposed to different oxygen concentrations. Scale bar: 40 µm. (**e**) Confocal imaging of GFP signal of *HRPE-Ω:GG* root tips at 21% or 0% O_2_. (**f**) Histochemical GUS staining of *HRPE-Ω:GG* at 21 and 0% O_2_. (**g**) RT-qPCR analysis of *GFP*, *GUS*, and *PCO1* expression in *HRPE-Ω:GG* seedlings exposed to 21 and 0% O_2_. *UBQ10* was used as a housekeeping gene. The expression level of each gene was calculated relative to its expression at 21% O_2_. Hypoxia treatments were performed overnight. Anoxia treatment was performed for 4 h. Two-sided t-test. Stars indicate a statistical significant difference (**** *p* value < 0.0001).

**Figure 3 biosensors-10-00197-f003:**
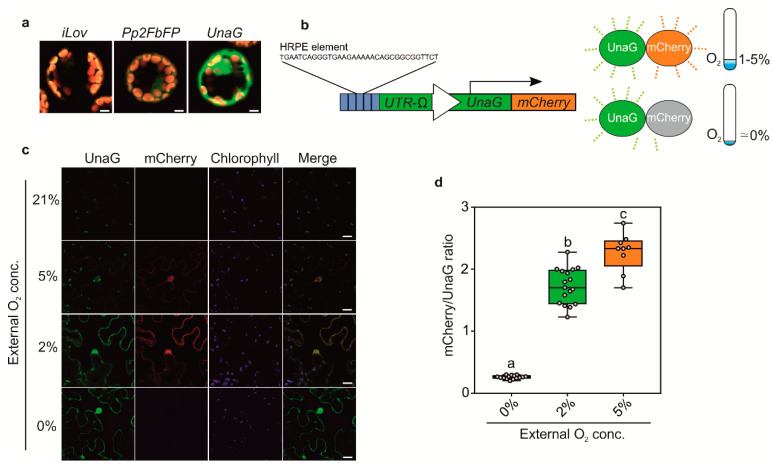
(**a**) Confocal microscopy analysis of iLOV, Pp2FbFP, and UnaG overexpressed using a *35S:* promoter in *Arabidopsis thaliana* protoplasts. Scale bars 5 µm. (**b**) Schematic representation of the *HRPE-Ω:UnaG-mCherry* sensor. (**c**) Confocal microscopy analysis of *HRPE-Ω:UnaG-mCherry* in the epidermis of *Nicotiana benthamiana* leaves exposed to 21%, 5%, 2%, and 0% O_2_. Scale bars 20 µm. (**d**) Boxplot representation of mCherry/UnaG ratios at 5%, 2%, and 0% O_2_ as imaged in c. One-way analysis of variance (ANOVA) followed by Tukey post hoc test. Letters indicate a statistical significant difference (*p* value < 0.05).

**Figure 4 biosensors-10-00197-f004:**
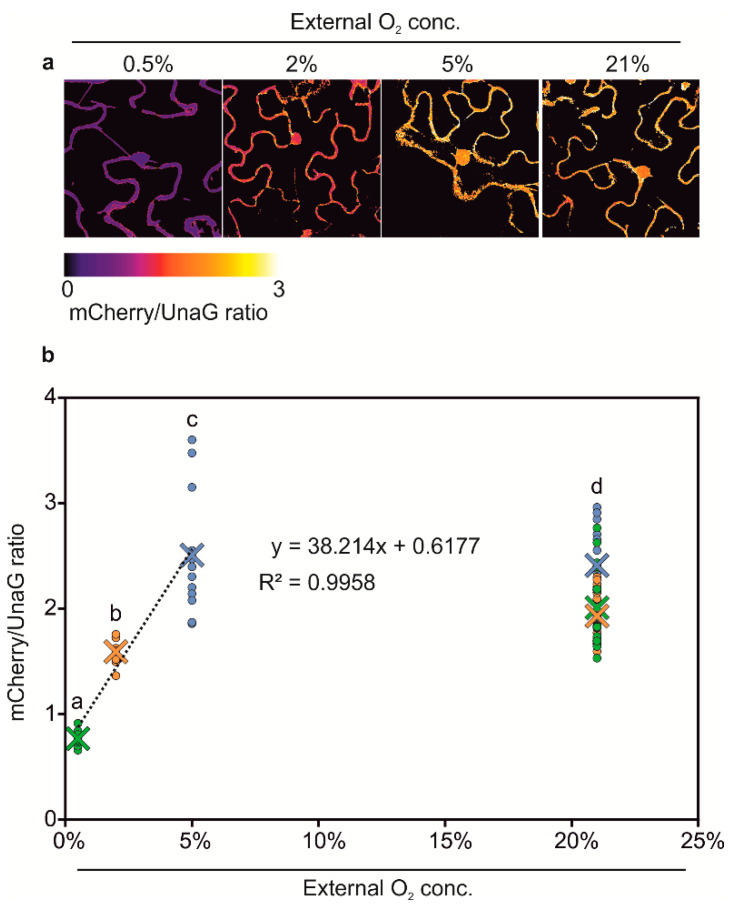
(**a**) Confocal microscopy analysis of *pUBQ10:UnaG-mCherry* in transiently transfected *Nicotiana benthamiana* leaves after treatment with various O_2_ concentrations. mCherry maturation is O_2_-dependent, while UnaG is not. Images display the mCherry/UnaG ratio, corresponding to the tissue O_2_ concentration. (**b**) Quantification of mCherry/UnaG ratios at different O_2_ concentrations. The green, orange, and blue data points represent the 21% O_2_ control for 0.5%, 2%, and 5% O_2_, respectively. Crosses represent the means. The linear regression was calculated using the ratios at 0.5%, 2%, and 5% O_2_. Two-way analysis of variance (ANOVA) followed by Tukey post hoc test. Letters indicate a statistical significant difference (*p* value < 0.05).

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
