# Peer review of "An Improved HRPE-Based Transcriptional Output Reporter to Detect Hypoxia and Anoxia in Plant Tissue"

_biosensors, 2020, doi:10.3390/bios10120197_

Round 1

Reviewer 1 Report

This report by Panicucci et al is focussed on the development and investigation of hypoxia reporter constructs for detecting variation in oxygen-levels in plant cells.  Hypoxia occurs frequently in plants, both acutely (in response to abiotic and biotic stresses), but also chronically in certain tissues and regions (e.g. due to diffusion limitation and high levels of energy consumption).  As such, the development of improved reporters for monitoring tissue oxygen concentrations is a worthwhile endeavour.  Overall this was a well written manuscript, with clear and appropriate figures.  I have several comments that I would like the authors to address or discuss further:

  • I have a query regarding the use of Luciferase (both Firefly and Renilla) to monitor promoter/UTR activity under hypoxia in figure 1. It is my understanding that the catalysis of luciferin by luciferases requires molecular oxygen.  Therefore, could the authors comment on the efficacy/appropriateness of using luciferase as a reporter for monitoring these promoter activities under reduced oxygen levels?
  • In figure 1, reporter activity is most significantly enhanced after 4h hypoxia, which seems to be quite a long time. Can the authors provide extra commentary on how this might limit the use of such a reporter when attempting to measure acute or rapid-onset hypoxia in response to a stimulus or stress? (I appreciate that there is partial discussion of reporter limitations in the discussion)
  • In figure 2B – why could the HRPE-GUS not be observed in the shoot apex at 21% O2? This region is normally hypoxic even under exogenous normoxia, and If I recall such a pattern was observed for this reporter in Weits et al 2019 Nature?
  • Authors clearly show that anoxia impairs GFP/GUS reporter function, which is likely due to a requirement of GFP maturation for molecular oxygen. Does this mean therefore that even when GFP is observed under hypoxia, the signal may not be a true reflection of the actual hypoxic state?  What I mean is, under severe hypoxia, although oxygen is still present in sufficient amounts for SOME GFP maturation, its levels would be rate limiting and the full extent of the reporter’s activity is likely somewhat inhibited.  Could authors discuss this scenario further?
  • I like the idea or a ratiometirc anoxia/hypoxia sensor, whereby the ratio of UnaG to mCherry can be used to distinguish anoxia from hypoxia, and the authors clearly show that mCherry signal is absent under anoxia whilst UnaG persists. I wonder if it would be possible for the authors to examine this reporter under an incremental range of O2 concentrations, and use it to generate a standardised list of UnaG:mCherry ratios?  This could then be used as a standard for cross referencing ratio observations in a range of plant tissues with known concentrations of oxygen.  Is this something the authors think is feasible?

Minor points:

  • Line 47 should say “than” not “then”
  • Line 87 should cite Licausi et al (2011) and Gibbs et al (2011) Nature.
  • Figure 2c is not cross-referenced in the main text.
  • References 4-44 are cited int ext but missing from the reference list (which only goes up to 39). Please add the missing references to the reference list.

Reviewer 2 Report

The manuscript "An improved HRPE-based transcriptional output reporter to detect hypoxia and anoxia in plant tissue" submitted to Biosensors has been reviewed. The authors designed and optimized a hypoxia-signaling reporter by testing the effects of using different UTRs. The authors also compared their findings with traditional and explained advantages.

Overall it is in good shape regarding to how the authors tell the story step-by-step. I enjoyed reading it. Great job. Please see attached for few comments.

Author Response

We are grateful to the reviewer for the examination of our manuscript. We are delighted that you enjoyed reading it! 

A description of the material harvested and treatments performed for the qPCR analysis is now provided in the material and methods [lines 179-181]

We also clarifed in Figure legend 2 how the relative expression of each gene was calculated [lines 248-249]. We used UBQ10 as housekeeping gene and compared the expression of each gene to its expression at the 21% oxygen control treatment. 

4-weeks was replaced with four-weeks [Line 196]

Reviewer 3 Report

This manuscript presents the optimisation of a secondary hypoxia reporter - secondary in that it relies on perception via hypoxia responsive transcription factors (as noted by the authors in the discussion). While the central promoter element (HRPE) has been characterised previously, it's practical application had been questioned due to the oxygen-dependence of both translation and GFP fluoresence. This manuscript firstly tailored the HRPE for improved translation at low oxygen availability by incorporating a UTR from the 35S promoter or TMV. They have then demonstrated that GFP fluorescence is oxygen-limited in planta but apparently not because of the oxygen dependence of fluoresence per se, rather due to the dependence of translation, as GUS was also imprecise. Nevertheless, the authors demonstrated the reliable use of the O2-independent UnaG chromophore. The concluding reporter was based on the optimised HRPE driving a UnaG-mCherry fusion protein. 

My criticisms of this manscript are few, as it is elegant and thorough. I have made comments on the pdf attached. 

Author Response

We are grateful to the reviewer for the detailled assessment of our manuscript. 

We clarified the treatment performed by Iacopino and colleagues [line 63]

A sentence was added to the introduction to highlight that the original GFP based reporters required improvements due to the requirement of oxygen for GFP maturation [line 70-71]. 

A description of the stars was added to figure legend 1. [line 231]

The remaining spellings spotted by the reviewer have been corrected and are highlighted using a red font. 

Round 2

Reviewer 1 Report

I am satisfied with the authors responses to my queries.  I believe this paper is ready to be published and will be of interest to many in the field.